# Cross-Correlation- and Entropy-Based Measures of Movement Synchrony: Non-Convergence of Measures Leads to Different Associations with Depressive Symptoms

**DOI:** 10.3390/e24091307

**Published:** 2022-09-15

**Authors:** Uwe Altmann, Bernhard Strauss, Wolfgang Tschacher

**Affiliations:** 1Institute of Psychosocial Medicine, Psychotherapy and Psycho-Oncology, Jena University Hospital, D-07743 Jena, Germany; 2Department of Experimental Psychology, University Hospital of Psychiatry and Psychotherapy, CH-3060 Bern, Switzerland

**Keywords:** nonverbal communication, movement synchrony, time-series analysis, validity, depression

## Abstract

Background: Several algorithms have been proposed to quantify synchronization. However, little is known about their convergent and predictive validity. Methods: The sample included 30 persons who completed a manualized interview focusing on psychosomatic symptoms. The intensity of body motions was measured using motion-energy analysis. We computed several measures of movement synchrony based on the time series of the interviewer and participant: mutual information, windowed cross-recurrence analysis, cross-correlation, rMEA, SUSY, SUCO, WCLC–PP and WCLR–PP. Depressive symptoms were assessed with the Patient Health Questionnaire (PHQ9). Results: According to the explorative factor analyses, all the variants of cross-correlation and all the measures of SUSY, SUCO and rMEA–WCC led to similar synchrony measures and could be assigned to the same factor. All the mutual-information measures, rMEA–WCLC, WCLC–PP–F, WCLC–PP–R2, WCLR–PP–F, and WinCRQA–DET loaded on the second factor. Depressive symptoms correlated negatively with WCLC–PP–F and WCLR–PP–F and positively with rMEA–WCC, SUCO–ES–CO, and MI–Z. Conclusion: More standardization efforts are needed because different synchrony measures have little convergent validity, which can lead to contradictory conclusions concerning associations between depressive symptoms and movement synchrony using the same dataset.

## 1. Introduction

Processes relevant in psychotherapy can be located on different time scales ranging from neuronal processes that change within milliseconds, to affective and interpersonal processes representing single sessions, to between-session changes of mood stages [1,2,3]. Both bottom-up effects (e.g., when patient–therapist interactions have impacts on patient’s mood) and top-down effects (e.g., mood affecting the kind of interpersonal interaction) are assumed [1]. The core of psychotherapy process is generally considered to rest in the exchanges between the patient and therapist, which consist of verbal–semantic and nonverbal aspects.

The nonverbal interaction of patient and therapist may be understood as a coupled dynamical system [4,5,6,7,8]. Each sub-system (the patient’s or the therapist’s) obeys its own eigen-dynamics and coupling dynamics. The eigen-dynamic is constituted by an actor’s ability to perceive and process information and act accordingly (see Figure 1 left). The coupling dynamic refers to the mutual influence between the patient and therapist, may be asymmetrical (e.g., the therapist affecting the patient’s state more than vice versa) and may change during the interpersonal interaction (e.g., at the beginning of a session, the patient influences the therapist, whereas at the end the influence is reversed). There are two different understandings of coupling dynamics. One is that the degree of coupling changes more or less smoothly over time [9,10,11], the other regards the coupling dynamics as an on–off process whereby phases with no or very weak coupling (no synchronization visible) may alternate with strongly coupled phases [5,8]. The former dynamics may be assumed in more stationary processes (e.g., physiological data), whereas the latter on–off dynamics occur in behavioral processes with non-stationary bursts (e.g., movement activity). Phases of strong coupling are characterized by a synchronization of sub-system states and are hence called synchronization intervals [4,12] or mimicry episodes [13]. The person who acts as the driver during the coupling is called leader, and the person who follows/imitates is the driven (see Figure 1 right).

### 1.1. Synchronization in Patient–Therapist Interaction

Psychotherapy research has investigated the synchronization of physiological parameters [14,15], body movements [5,11], facial expressions [16], prosodic cues [17,18,19], and language style [20,21]. Many studies have investigated the relationship between (nonverbal) synchronization and therapeutic success. According to the systematic review of Wiltshire et al. [22], physiological synchrony was most frequently correlated with empathy and language, vocal synchrony with therapeutic alliance, and movement synchrony with psychotherapy outcomes. This review supported the InSync model of psychotherapy [3,23], which assumes that (nonverbal) synchrony in patient–therapist interaction affects the emotion regulation of patients (as a top-down effect at medium/tonic to longer/chronic time scales) as well as the therapeutic relationship and, as a consequence, therapeutic success (bottom-up effect at tonic and fast/phasic time scales).

Other psychotherapy studies have investigated (nonverbal) synchrony in interpersonal interaction as a diagnostic feature of mental disorders. Multiple studies suggested, for example, that attenuated nonverbal synchrony was linked with depressive symptoms [16,24,25,26]. These findings correspond with neurophysiological [27,28] and interpersonal models of depression [29]. The former explains changes in emotion regulation and interpersonal interaction (e.g., less smiling or movements) by disorder-related changes in neurophysiological processes (e.g., dysfunctions in the left frontal hemisphere of the brain) [30,31]. Interpersonal models of depression [29] assume that depressed persons induce a negative mood in their conversation partners, thus provoking negative responses, which in turn confirm the negative expectations of the depressed person. Accordingly, in interviews with depressed patients, the interviewers synchronize their nonverbal behavior less often.

It may be noted, however, that the findings are not homogeneous. Some studies did not find significant associations between synchrony and therapeutic outcome [17,32], or even reported that higher synchrony was related to higher symptom levels [33,34]. This was also true for synchrony as a diagnostic feature, e.g., when more vocal synchrony was correlated with higher anxiety symptoms in the study of [35].

From a methodological point of view, an explanation of the conflicting results may be that nonverbal synchrony was measured differently. Thus, researchers may have addressed different aspects, or even different concepts, of synchrony, which may have resulted in varying correlations between synchrony and therapeutic outcome as well as symptom load [36]. This unsettled state of research has motivated the present study on the validity of different synchronization measures.

### 1.2. Measures of Synchronization and Its Convergent Validity

Various statistical methods may be used to estimate the average degree of coupling (e.g., [14,37]) or identify synchronization intervals (e.g., [12]). Despite multiple overviews of methods applied in psychotherapy research [10,36,38] so far, there are only few studies on the validity of synchrony measures.

First, it should be noted that synchrony measures depend on the parameter settings of algorithms. Ramseyer and Tschacher [39], Schoenherr et al. [40] and Behrens et al. [41] applied windowed cross-lagged correlation algorithms multiple times to the same bivariate time series and varied parameters such as degree of smoothing, window size and maximum time lag. Among other things, they showed that smaller windows [39,40,41], non-transformed movement data and slight smoothing [40] lead to higher synchronization values and higher correlations between synchrony and therapeutic alliance, respectively [39]. The application of a pseudo-synchrony approach [42] also affects the measured synchrony. For each real-world time-series pair, Moulder et al. [43] generated multiple artificial time-series pairs by (a) shuffling participants, (b) shuffling time-series segments within a dyad, (c) shuffling measurement points within a dyad and (d) reversing one of the time series in the pair. They found that the decision as to whether synchrony was present in a time-series pair strongly depended on the applied shuffling method. All these findings imply that the validity of synchrony measures depends on the parameter settings of an algorithm.

Regarding the validity of synchrony measures, one should distinguish different kinds of validity. Predictive validity is present when a synchrony measure predicts an external criterion in accordance with theoretical expectations, as was investigated by [36,39,44,45]. Feniger-Schaal, Schoenherr, Altmann and Strauss [44] applied windowed cross-lagged correlation (WCLC) with peak picking (PP) by [12] to motion time series that were captured in a “mirror game”. In the first phase of the mirror game, the study assistant mirrored the movements of a participant. In the second phase, the leading–following roles were switched. In the third phase, these roles were not predetermined. In concordance with instructions, the algorithm measured more synchrony with the participant leading in the first study phase, and more synchrony with the assistant leading in the second phase. In the study of Luehof [45], WCLC with PP by [9], windowed cross-lagged regression (WCLR) with PP by [4,12], and recurrence quantification analysis (RQA) were able to discriminate between interviews with rapport-trained interviewers and control interviewers, finding more synchrony with the trained interviewers. The WCLR–PP showed the best discrimination. Schoenherr et al. [36] used therapeutic success as the criterion to be predicted by synchrony. They found that only windowed cross-correlation (WCC), WCLC–PP and WCLR–PP correlated significantly in the expected direction with therapeutic success.

Schoenherr, Paulick, Deisenhofer, Schwartz, Rubel, Lutz, Strauss and Altmann [40] studied the criterion validity of synchronization measures using artificially generated time-series pairs that contained a single synchronization interval. The WCLC–PP and WCLR–PP by [4,12] were applied to each time-series pair and correct identifications of the synchronization interval (the criterion) were counted. The best concordance in terms of the average Cohen’s κ was observed for both WCLC–PP and WCLR–PP with window widths of 3 and 5 s, non-transformed movement data and slight smoothing. When applying the algorithms to real motion time series with isolated synchronization intervals (no other movement activity before or after the synchronization interval), the identification rate varied between moderate and substantial Cohen’s κ, depending on the parameter settings.

Validity defined as congruence between different measures was investigated by Schoenherr, Paulick, Worrack, Strauss, Rubel, Schwartz, Deisenhofer, Lutz and Altmann [36], Luehof [45] and Tschacher and Meier [14]. They applied multiple algorithms to naturalistic bivariate time series and determined convergent validity by the correlations between different synchrony measures. In their study of physiological synchrony, Tschacher and Meier [14] found little or no inter-correlations between SUSY–ES_abs_, SUSY–ES_noabs_ and the SUCO algorithm. In a study of movement synchrony, Luehof [45] found no concordance between the synchrony measures of WCLC–PP by [9], WCLR–PP by [4,12], and recurrence quantification analysis (RQA). In the study of Schoenherr, Paulick, Worrack, Strauss, Rubel, Schwartz, Deisenhofer, Lutz and Altmann [36], cross-lagged correlation (CLC), cross-lagged regression (CLR), windowed cross-correlation (WCC), windowed cross-lagged correlation (WCLC) by [37], WCLC by [32], WCLC–PP and WCLR–PP by [4,12], and cross-recurrence quantification analysis (CRQA) by [46] were conducted. The correlation between two synchrony measures ranged from not present (*r* ≈ 0) to almost perfect (*r* ≈ 1). In a subsequent exploratory factor analysis, CLC, WCLC by [37], and WCLC by [32] formed a factor of highly correlated synchrony measures. The second factor loaded on average cross-correlation within the synchronization intervals assessed with WCLC–PP and WCLR–PP by [12]. The third factor consisted of non-linear synchrony such as CRQA and the frequency of synchrony of WCLC–PP and WCLR–PP by [12]. Schoenherr et al. [36] concluded that the examined algorithms did not measure the same kind of synchrony and that different measures predicted different effects on therapeutic outcome.

### 1.3. Research Question

The present study explored the convergent validity and predictive validity of cross-correlation- and entropy-based measures of movement synchrony. We used data from a pilot study on nonverbal communication in depressive patients and healthy controls [16,47,48]. The primary study focused on the evaluation of feasibility of recruitment, assessment procedures, automatic coding of nonverbal behavior and provided first empirical results on the differences between patients with depression and healthy controls in terms of body motion, facial expressions and prosody. In the present secondary analysis, we addressed a methodological question: the validity of movement synchronization measures. For this purpose, multiple algorithms measuring synchronization were applied to motion times series of participants and interviewers. The convergent validity was examined by correlations between the synchrony measures. According to [36], we assume weak convergent validity in terms of low correlations between different synchrony measures, and that some measures can be assigned to different facets of synchrony measures. As in [36], we conducted an exploratory factor analysis to identify the facets of synchronization. Due to the fact that the distribution of synchrony measures is non-normal, we conducted a minimum rank factor analysis, which is more appropriate for non-normally distributed data. The predictive validity was investigated by comparing the synchrony measures in patients with major depression and in healthy controls as well as by the correlation between synchrony measures and symptom load, which was assessed with questionnaires. According to the literature mentioned above, movement synchrony was expected to be lower in the interview dyads with depressive patients.

To our knowledge, the present study is the second peer-reviewed study on the convergent validity of synchronization measures applied in clinical research. In the first study [36], the synchronization of the patient and psychotherapist in an early therapy session was investigated. In comparison to [36], we applied additional synchronization measures, especially measures based on information theory, and the homogeneity of the interactions was given (manual-guided interviews rather than therapy sessions addressing patient-specific conversation topics in [36]), and predicted criterion and synchrony were much closer in terms of time (the criterion—depression—was assessed before the interviews rather than measured weeks after the sessions, as was true for the criterion—reduction of interpersonal problems—in [36]).

## 2. Materials and Methods

### 2.1. Sample of Participants

The sample included 15 inpatients with major depression and 15 healthy controls matched by age and gender, thus groups did not differ regarding mean age and gender distribution. The age range was 20 to 30 years. Of the 30 participants 40% were female. Table 1 gives a short description of both groups, which showed no group differences regarding further socio-demographic characteristics. Patients with depression reported higher degrees of depressive and anxiety symptoms. For a detailed description of inclusion criteria, recruitment, and group characteristics, see the primary study [16].

### 2.2. Instruments

Prior to the interviews, several questionnaires were administered. We assessed the degree of depressive symptoms using the corresponding scale of the Patient Health Questionnaire (PHQ9) [49]. A sum score of 0–4 is interpreted as no or minimal, 5–9 as mild, 10–14 as moderate, and 15–27 as severe depressive symptoms. The degree of anxiety symptoms was measured with the Generalized Anxiety Disorder Scale (GAD7) [49]. The values 0–4 are interpreted as no or minimal, 5–9 as mild, 10–14 as moderate, and 15–21 as severe anxiety symptoms. Both sum scores have an acceptable internal consistency (Cronbach’s α > 0.8). Further questionnaires were assessed in the primary study but not used in the present study.

### 2.3. Interviews of Patients and Controls

The focus of the interviews was on somatic complaints, which may be present in healthy participants as well, similar to anamnestic interviews regarding somatoform disorders (SCID-I, section G) [50]. Example questions were: “Have you been ill during the last few years?”, or “Have you had any significant problems with headaches?”. At the beginning of interviews, the interviewer asked warm-up questions (e.g., “Did you find your way to the site easily?”) to allow the interviewee to become accustomed to the recording situation (cameras, etc.).

We used two cameras to record a frontal view of each person. Both recordings were subsequently synchronized by means of a film clapperboard and merged into a split-screen video. Interviews were held in a neutral counseling room where the interviewee and interviewer sat across each other at a table on identical chairs. The interviews were conducted by two female medical students (age ~25 years) in their senior semester. Both interviewers were trained and instructed to adopt a professional and neutral style. Further details on interviews and video recording are reported in [16,48].

### 2.4. Coding of Motor Activity during the Interviews

Using the interview videos, the motor activity of the interviewees and interviewers was captured using motion-energy analysis, or MEA [51]. We applied the MATLAB© scripts developed by Altmann [4,12], where regions of interest (ROI) can be drawn by hand [5] (free download at https://github.com/10101-00001/MEA, accessed on 15 July 2022). To capture motor activity, the MEA considers all changes of subsequent (*t*; *t* + 1) video frames of the recording. First, for each person, a ROI is defined that covers the region in which movements are visible. Inside the ROI, those pixels are counted whose grayscale values change substantially from *t* to *t* + 1 (cut-off value: 12 of 256 grayscale degrees). The number of such pixels defines the motion energy of the respective person’s ROI at *t*. For each of the 30 interview videos, we thus generated a bivariate time series that represents in detail (25 measures per second) the visible movement activity (movement of torso, arm, hands, and head of each interlocutor were aggregated to one measure of individual motion energy).

After the MEA, we applied several pre-processing steps. First, each time series was standardized by the size of the corresponding ROI and multiplied by 100. Accordingly, the values of time series ranged from zero (no motion) to 100 (entire ROI activated). Finally, all the time series were smoothed using a moving median with a bandwidth of five frames.

Figure 2 shows, as an example, one pair of motion-energy time series to which the aforementioned preprocessing steps were applied. The time axis is given in frames (25 frames = 1 s). Some algorithms analyze the time series window-wise, e.g., in rMEA und WinCRQA, 1500 frames, or in WCLR–PP, 125 frames. To illustrate what amount of motion activity was captured during such intervals, in Figure 2 we plotted examples of time series segments with length 1500 frames and 125 frames, respectively.

The length of the time series ranged from 10,325 to 42,250 frames (from 413 to 1850 s, respectively; median = 855 s). The interviews of the patients lasted longer than those of controls (median_Patients_ = 1276 s, median_Controls_ = 759 s, median test *p* = 0.001).

### 2.5. Measures of Movement Synchrony

The 30 bivariate time series originating from the interviews served as the data input for the synchronization measures that we wished to assess. Features of the measures provided by the algorithms introduced below are summarized in Table 2.

#### 2.5.1. Cross-Correlation

The “simplest” measure of movement synchrony is the cross-correlation (CC) of both time series describing the movements of interlocutors. Please note that in this approach no time lag between both time series is modeled.

Some research has considered the sign problem when the averages of cross-correlations are computed: For example, a time series may include sections with *r* = 0.5 and the same number of sections with *r* = −0.5, so that the aggregated cross-correlation is zero, leading to the conclusion that on average there is no interrelatedness, or no synchrony. To avoid this problem, prior to aggregation, the signs of cross-correlations may be removed by calculating the absolute values (e.g., [37,42,55]), or the coefficient of determination (squaring the correlations) may be used (e.g., [12]). In the latter approach, large cross-correlations will be weighted higher. Furthermore, sometimes Fisher’s Z-transformed correlation is considered, because then values are approximately normally distributed. Since a consensus has not been reached, we considered all the options in the present study: raw values of cross-correlations (including negative and positive values when averaging; CC–raw), the absolute values (CC–abs), Fisher’s Z-transformed (CC–Z), and squaring of cross-correlations (CC–R2).

#### 2.5.2. rMEA

The R package rMEA [37,52] is based on the work of Ramseyer and Tschacher [11,42] and includes motion capture via MEA as well as (a variant of) windowed cross-lagged correlation (WCLC) to compute the averages of local cross-correlations. Similar to the approach of Boker, Rotondo, Xu and King [9], local associations of both time series are quantified by the cross-lagged correlations of time-series segments—so-called windows. When starting the algorithm, the user has to define the window size (default value: *b* = 60 s, i.e. 60·25=1500 respective time points when the video frame rate per second is 25) as well as the maximum time lag (default value: $\tau_{max}=5\;s$, 125 respective time points) which defines the range of the considered time lags. The calculation of WCLC contains three steps. First, a cross-correlation (time lag *τ* = 0) for a pair of windows with the same starting point (*t*) is computed. Second, the start position of the reference window is kept constant, whereas the start position of the second window is shifted up to the maximum time lag. In the third phase, the position of the reference window is shifted with an increment of 30 s (default value). Then, the algorithm repeats step 1 and 2 for this start position of the reference window. The result is a matrix whose columns refer to the time lag ($-\tau_{max},\ldots ,\;0,\;\ldots ,\tau_{max}$) and rows to the start position of the reference window (1,…,L−b+1; *L*: time-series length, *b*: window width). The values of this matrix are the Fisher’s Z-transformed coefficients of WCLC (default setting). Before applying the transformation, the absolute values of all cross-correlations are computed to remove the signs (default setting).

In the present study we considered two measures of the degree of synchrony provided by the rMEA package: First, the average windowed cross-correlations (step 1 above; rMEA–WCC), and second, the average windowed cross-lagged correlations (step 3 above; rMEA–WCLC). The former only includes values of the column of the WCLC matrix referring to *τ* = 0, whereas the latter considers all columns. In contrast to Boker, Rotondo, Xu and King [9] and Altmann [4,12] there is no selection of WCLC maxima, thus no application of a peak-picking algorithm.

Due to the fact that noise and non-stationarity can cause cross-lagged correlation, a pseudo-synchrony approach [42,56] is conducted in the next step. The corresponding bootstrap algorithm randomly recombines the time series of person A and person B of another interview multiple times (default value: 100 times) and each time computes the WCLC. Thus, the surrogate generation is based on person shuffling. In this way, a statistic is produced to test whether the present WCLCs are different from the expected value of a random distribution of WCLC values.

#### 2.5.3. SUSY

Surrogate synchrony (SUSY) [53,55] is based on the cross-correlation function of dyadic time series (the algorithm can be used online: https://www.embodiment.ch/, accessed on 15 July 2022). The cross-correlations are computed across a range of lags *L* (here −5 s ≤ *L* ≤ 5 s). The cross-correlation function is computed segment-wise, i.e., separately in all non-overlapping segments of the time series (here segment-size = 30 s). It may be noted that terminology differs in the rMEA package, where “window” is used to denote segments. All cross-correlations are transformed using Fisher’s *Z*-transformation to allow the aggregation of cross-correlations. The synchrony of any segment is then defined as the mean of all (lagged) cross-correlations of this segment, and the synchrony of the time series as the mean of segment means. Aggregation is performed using either absolute *Z* cross-correlations (*Z*abs) or the original, negative or positive, cross-correlations (*Z*noabs). The reason for taking the absolutes of correlations is that one may define synchrony irrespective of the direction of coupling, which may be negative (“anti-phase”) or positive (“in-phase”); in *Z*abs, both are collapsed into one signature of synchrony. *Z*noabs differentiates in-phase from anti-phase coupling. Then surrogate tests are performed to establish a control condition for the *Z*abs and *Z*noabs values of each dyad. Surrogate time series in SUSY are generated by randomly shuffling the sequence of segments, independently for each dyad member (surrogate generation by segment shuffling). From a dyadic time series with *n* segments, *n*(*n* − 1) surrogates can be produced. In the present analysis, all *n*(*n* − 1) respective surrogates were used. The surrogate step produces effect sizes (ES) as the final signatures of synchrony in SUSY, namely *SUSY–ES*_abs_ and *SUSY–ES*_noabs_. *SUSY–ES*_abs_ is derived using the mean surrogate *Z*abs and their standard distribution: *SUSY–ES*_abs_ = (*Z*abs − *Z*abs-surr)/SD(*Z*abs-surr). *SUSY–ES*_noabs_ is computed analogously. The leading–following relationships of synchrony may further be operationalized by differentiating between positive and negative lags *L*.

#### 2.5.4. SUCO

Surrogate concordance (SUCO; online access https://www.embodiment.ch/, accessed on 15 July 2022) [14,55] is based on the correlations of the local slopes of dyadic time series (A,B). The slopes are determined by least-squares regression in windows (here, window size was 3 s) of the time series, and the time series are again partitioned into segments of 30 s duration as in SUSY. The linear slopes are computed inside the first window of segment *i*, the window is then shifted by an increment of 1 s and the slopes are again computed; thus, overlapping windows are used. This is repeated until all windows in segment *i* are covered. The slopes in segment *i* of time series A are Pearson-correlated with those of the same segment of B. The resulting correlation *r**i* denotes the relation between A’s and B’s slopes in segment *i* of the time series. This is performed in all segments of the time series A and B. The absolute values of all correlations *r**i* are Fisher’s *Z*-transformed and aggregated, yielding *Z’*abs (with high comma to distinguish from SUSY). Segment-wise shuffling is used, as in SUSY, to create surrogate time series, yielding the effect size of *Z’*abs, labeled *SUCO–ES*_abs_. The concordance index (*SUCO–CO*) across all segments of the client–therapist interaction is defined by the natural logarithm of the sum of all positive correlations *r**i* divided by the absolute value of the sum of all negative correlations *r**i*, as previously suggested by Marci and Orr [57]. Using surrogate analysis, an effect size *SUCO–ES–CO* is computed by standardizing the concordance index by the mean and standard deviation of the concordance indexes of surrogate data, in analogy to the procedure in SUSY. The leading–following relationships of concordance synchrony are operationalized by shifting of A’s windows with respect to B prior to computing *r**i*, yet such lags were not computed in the present analyses.

#### 2.5.5. WCLC–PP and WCLR–PP

The algorithm by Altmann [4,12] (download at https://github.com/10101-00001/sync_ident, accessed on 15 July 2022) combines three approaches: First, the computation of local associations proposed by Boker, Rotondo, Xu and King [9], Ramseyer and Tschacher [11] and Watanabe [58]; second, the reduction of auto-correlation bias [59] by a regression approach, e.g., as performed by Gottman and Ringland [60]; and third, the differentiation between significant and non-significant local associations and their selection by a peak-picking algorithm as proposed by Boker, Rotondo, Xu and King [9]. The algorithm is based on the assumptions that in interpersonal interaction, phases of synchrony (high degree of coupling) alternate with phases of non-synchrony (no coupling), and that within a phase of synchrony the data are sufficiently described by cross-lagged regression. The validity and high detection rate of the algorithm has been shown in multiple studies [36,40,44,45].

The computation consists of three steps. First, the local associations are computed. This can be performed with windowed cross-lagged correlation (WCLC) or windowed cross-lagged regression (WCLR). Similar to rMEA and SUSY, time-lagged windows of both time series are considered. In WCLR, for each start position of a reference window (e.g., of person A) and each possible time lag (*τ*), two cross-lagged regressions are applied. In model 1, the window of person A beginning at *t* + *τ* is predicted by the window of person A beginning at *t*. This means that only the auto-correlation is modeled. However, in model 2, the window of person A beginning at *t* + *τ* is predicted by a window of person A beginning at *t* (corresponding to the auto-correlation) as well as a window of person B beginning at *t* (corresponding to the cross-correlation). Then, the coefficient of determination is computed based on both models: $\Delta R_{t,\tau }^{2}=R_{M2,t,\tau }^{2}-R_{M1,t,\tau }^{2}$ (note: M1: model 1; M2: model 2; *t*: start position of window; *τ*: time lag between “action” and “response”). $\Delta R_{t,\tau }^{2}$ quantifies the proportion of variance of window A at *t*, which is explained by cross-lagged correlation with time lag *τ* and which is unbiased by the auto-correlation with time lag *τ*. The procedure described above is conducted for each position of the reference window (t∈{1,…,L−b+1}; *L*: length of time series; *b*: window width) and each time lag of interest ($\tau \in \left\{-\tau_{max},\ldots ,\tau_{max}\right\}$). All resulting $\Delta R_{t,\tau }^{2}$ values are stored in a matrix (so-called R square matrix; [12]). Similar to rMEA, the column refers to the time lag ($-\tau_{max},\ldots ,\;0,\;\ldots ,\tau_{max}$) and the row to the start position of the reference window (1,…,L−b+1; *L*: time-series length; *b*: window width). However, the values of matrix ($\Delta R_{t,\tau }^{2}$) are not correlation coefficients but the proportion of explained variance by cross-correlation adjusted by auto-correlation with the same time lag (also called R square or coefficient of determination). In contrast to WCLR, the WCLC by Altmann [4,12] estimates the local associations between two time-series windows with cross-lagged correlations. They can be confounded by auto-correlation. However, the process is similar: The correlations computed for windows that are time-lagged and “moved” over the time axis. The result is also a R square matrix. Its elements ($R_{t,\tau }^{2}$) are the squared windowed cross-lagged correlations at a specific start position of reference window (*t*) and a specific time lag (*τ*) of the interlocutor’s window.

In the second step of the analysis, the R square matrix is analyzed by a peak-picking algorithm (abbreviation: PP) [4,12]. For each start position of reference window at *t*, local maxima of $\Delta R_{t,\tau }^{2}$ ($R_{t,\tau }^{2}$) values are detected (for illustrations see [12]). Next, local maxima with equal time lag and directly consecutive in time are combined into intervals. When a start position of the reference window is part of multiple intervals, then the interval with the largest average $\Delta R_{t,\tau }^{2}$ is selected. These selected intervals are called synchronization intervals [12]. The output of the peak-picking algorithm is a list of synchronization intervals (abbreviation: LOSI). Based on this list, an interpersonal interaction can be described as a process where phases of movement synchronization (synchronization intervals with a high degree of cross-lagged correlation) alternate with phases without synchronization (without significant cross-lagged correlation).

In the last step of WCLC–PP and WCLR–PP, various synchronization measures can be quantified based on the LOSI. In the present study, we considered the frequency of movement synchrony defined as the proportion of synchronization intervals in relation to the duration of the time-series length (WCLC–PP–F and WCLR–PP–F) and the average interrelatedness of both time series within the synchronization intervals quantified by the average R square of the synchronization intervals (WCLC–PP–R2 and WCLR–PP–R2).

Before starting WCLC–PP and WCLR–PP, some parameter values have to be defined. In the present study, the window width was 125 time points (5 s), the *R*^2^ cut-off was 0.25 (both values suggested by the simulation study of [40]), the increment was one frame (resulting in overlapping windows), and the maximum time lag *τ_max_* = 125 frames (both recommended by [5,8]). According to the simulation study of [40], in the LOSI we considered only synchronization intervals with $average\left(\Delta R^{2}\right)>0.25$, which led to better identification rates and lower false positives.

#### 2.5.6. Mutual Information

Mutual information (MI) [61] quantifies the amount of information that is shared by two random variables and uses this as a measure of dependence. Shannon information is closely linked with entropy [62]. In other terms, MI is the joint distribution of both time series related to the marginal distributions of both time series under the assumption of independence. In contrast to cross-correlation, which assumes continuous or interval-scaled time series, mutual information can only be computed for categorical variables or continuous variables binned into categories. A further difference is that MI does not rest on the assumption of linear dependencies between time series.

In the present study, we estimated MI using the R package mpmi (command cmi.pw) [54] which automatically calculates a vector of smoothing bandwidths for each of the dyadic time series. It uses a kernel-smoothing approach to estimate the joint distribution and both marginal distributions. The package provides three measures: an (uncorrected) raw value of MI (MI–raw), a Jackknife bias-corrected MI (MI–cor), and a Z-score of bias-corrected MI that provides a statistic for the null-hypothesis that the bias-corrected MI is zero (MI–Z).

#### 2.5.7. Recurrence Techniques

Cross-recurrence quantification analysis (CRQA) [63,64,65] is based on a state–space approach. Given time series of two coupled dynamical systems, in the first step, recurrence techniques identify the time points when both systems are in the same state (e.g., both interlocutors smile). This includes simultaneous and time-lagged states. The information is stored in the recurrence matrix (illustrated as a recurrence plot). In continuous data (e.g., movement intensity), a distance measure has to be defined (usually Euclidian distance) and a recurrence threshold (radius: ε) has to be specified. Instead of same states, the simultaneous and time-lagged similarity of continuous state parameters is identified ($\epsilon <||x_{t}-x_{t+\tau }||$).

The values of the recurrence matrix are zero or one, depending on the similarity of values (in categorical time series, identity of values) at a specific time point of the reference time series and time lag of the interlocutors’ time series. Based on the recurrence matrix, various parameters describing aspects of coupling can be computed, e.g., the percentage of recurrence points in the recurrence plot (recurrence rate: RR in %), the percentage of recurrence points forming diagonal lines (percentage of determinism: DET in %) or the Shannon information entropy of the diagonal line length longer than the minimum line (entropy: ENTR; entropy normalized by number of diagonal lines in the recurrence plot: rENTR) [46,66]. Of these measures, WinCRQA–DET is often used as a synchrony measure, e.g., in [40,67,68]. Please note that as in other algorithms, the result of the recurrence analysis depends on the parameter values, especially on the recurrence threshold [69].

In the present study, we conducted the windowed cross-recurrence quantification analysis (R command: wincrqa) implemented in the R package CRQA [46,66]. We transformed all the time series to the unit-interval and used a Euclidian distance with ε = 0.05 as recurrence threshold. The embedding dimension was three. As in rMEA, the window width was 1500 frames (60 s), the overlap of windows was 750 and the maximal time lag was 125 (5 s). The algorithms provided various outcome parameters for each window (e.g., RR, DET and ENTR). To obtain a measure for the entire conversation, we averaged these values over all the windows.

### 2.6. Statistical Analysis of Synchrony Measures

After the video recording of the 30 interviews and the measurement of motion energy during the interviews using the MEA, we applied the listed algorithms on the motion-energy time series to quantify synchrony. We created a data matrix in which a column refers to a specific synchrony measure and a line to an interview. Based on this table we investigated the validity of synchrony measures.

First, the convergent validity of synchronization measures was examined by Pearson and Spearman correlations. Thus, we assumed that all synchrony measures correlated with each other. According to Cohen [70], *r* > 0.1 can be interpreted as small, *r* > 0.3 as moderate and *r* > 0.5 as a large effect.

Due to the findings of [36], we explored facets of synchrony using factor analysis. To determine the number of extracted factors, we applied a parallel test with 100 bootstraps. We computed an exploratory factor analysis (EFA) with a maximum likelihood estimator (ML) as well as a minimum rank factor analysis (MINRANK), which is more appropriate in non-normally distributed data. In both factor analyses, the factors were allowed to correlate (oblimin rotation). An acceptable model fit is given when the root-mean-square error of approximation (RMSEA) is <0.08 and the Tucker Lewis Index (TLI) is >0.9.

Next, the predictive validity of synchronization measures was examined. We assumed that in dyads of patients with major depression, less movement synchronization would be observed than in the dyads of healthy controls. The synchronization measures of both groups were compared using the Kruskall–Wallis tests. In significant group differences, we reported Hedges *g* as an effect size measure. According to Cohen [70], *g* > 0.2 can be interpreted as small, *g* > 0.5 as moderate and *g* > 0.8 as a large effect.

Furthermore, for the predictive validity we assumed that a higher symptom load (assessed with PHQ9 and GAD7) would be related to less synchronization observed in the interviews. We computed Pearson and Spearman correlations.

## 3. Results

First, we investigated the convergent validity with correlations between different synchronization measures (Table 3). The three measures based on mutual information were highly interrelated (all Pearson *r* > 0.845, *p* < 0.01). This also holds for the three measures of the SUCO approach (all *r* > 0.718, all *p* < 0.05). Moderate correlations were found for both measures of rMEA (*r* = 0.685, *p* < 0.05), both measures of SUSY (*r* = 0.502, *p* < 0.05), both measures of WCLC–PP (*r* = 0.51, *p* < 0.05), and the four variants of cross-correlation (all *r* > 0.69, all *p* < 0.05).

Contrary to our expectation, no single synchrony measure significantly correlated with all other synchrony measures. Synchrony quantified as cross-correlation (CC–raw, CC–abs, CC–Z, and CC–R2) correlated with synchrony measures of SUSY, SUCO and rMEA package moderately (most Pearson *r* > 0.5). In contrast, the synchrony measures of WinCRQA correlated negatively with rMEA–WCLC (e.g., Pearson *r*(WinCRQA–RR, rMEA–WCLC) = −0.41, *p* < 0.05), WCLC–PP–F (e.g., *r*(WinCRQA–DET, WCLC–PP–F) = −0.49, *p* < 0.05), WCLR–PP–F (e.g., *r*(WinCRQA–DET, WCLR–PP–F) = −0.41, *p* < 0.05) and mutual information (e.g., *r*(WinCRQA–DET, MI–Z) = −0.43, *p* < 0.05).

The parallel test suggested for EFA and the minimum rank factor analysis that two factors best describe the considered movement-synchrony measures. The loadings of both factor analyses were similar (Table 4). The variants of cross-correlation, all measures of the SUSY package, all measures of the SUCO package, and the rMEA–WCC formed a factor. The indicators of the second factor were all variants of mutual information and rMEA–WCLC. In the minimum rank factor analysis, WCLC–PP–F, WCLC–PP–R2, WCLR–PP–F, and WinCRQA–DET were also assigned to the second factor. WCLR–PP–R2, WinCRQA–RR, and WinCRQA–ENTR had low loadings (<0.5) and were not assigned to either factor. rMEA–WCLC, WCLC–PP–F, and WCLR–PP–F showed large cross-loadings (>0.3). Accordingly, in both factor analyses the model fit described by RMSEA and TLI was not acceptable.

Next, we examined the predictive validity based on the criterion whether the synchronization measures predicted the assignment into the group of healthy controls or of depressed patients. Table 5 reports the group averages of the different synchronization measures as well as the *p*-value of group mean comparisons. When measuring synchrony with rMEA–WCC (*g*_Hedges_ = 0.838, *p* = 0.0274), SUCO–ES–CO (*g*_Hedges_ = 0.771, *p* = 0.0473), and MI–Z (*g*_Hedges_ = 0.882, *p* = 0.0197), we found that patients with depression had a higher degree of synchrony (in terms of interrelatedness) than the healthy controls. Such an association at a trend level was also found for SUSY–ES_abs_ (*g*_Hedges_ = 0.620, *p* = 0.0918) and SUCO–ES_abs_ (*g*_Hedges_ = 0.664, *p* = 0.0754). In contrast, WCLC–PP–F and WCLR–PP–F (measuring the frequency of synchronization intervals) indicated that patients with depression synchronized less often than healthy controls (WCLC–PP–F: *g*_Hedges_ = −1.03, *p* = 0.008 and WCLR–PP–F: *g*_Hedges_ = −0.994, *p* = 0.0114). All other synchrony measures were unrelated to group assignment.

To test predictive validity, we also examined the correlation between the degree of symptom load and synchronization measures (see Table 5). Similar to the group comparison, we found that rMEA–WCC (Spearman *r* = 0.49, *p* < 0.05), SUCO–ES_abs_ (*r* = 0.46, *p* < 0.05), SUCO–ES–CI (*r* = 0.49, *p* < 0.05), and MI–Z (*r* = 0.390, *p* < 0.05) correlated with the degree of depressive symptoms (PHQ9 sum-score) in terms of more depression leading to more synchrony. In contrast, the significant correlation coefficients of WCLC–PP–F (*r* = −0.43, *p* < 0.05) and WCLR–PP–F (*r* = −0.47, *p* < 0.05) suggested that more depression is related to less synchronization. Regarding the degree of anxiety (GAD7 sum-score), we found more significant correlations than for depressive symptoms (see Table 5). Many of these correlations between anxiety symptoms and synchrony were larger than the corresponding correlations between depressive symptoms and the synchrony measure (e.g., GAD7 and rMEA–CC: *r* = 0.600 versus PHQ9 and rMEA–CC: *r* = 0.46). In contrast, the correlation between the frequency measures of synchronization and anxiety symptoms were lower than the corresponding correlation between synchrony and depressive symptoms (e.g., GAD7 and WCLR–PP–F: *r* = −0.39 versus PHQ9 and WCLR–PP–F: *r* = −0.47).

## 4. Discussion

Nonverbal interpersonal interaction can be regarded as a complex dynamical system as it comprises a high number of elements, considers changes in time depending on external parameters, and may form temporary self-organized patterns that decrease the initially high entropy of these systems. One such pattern that has received considerable attention in recent social and clinical psychology is movement synchrony. Sequences of movement synchronization defined as temporally coordinated motor activity are characterized by a reduced degree of complexity and entropy, respectively, and a high degree of interrelatedness between participants and their behavior. Currently, several synchrony measures are available, some based on information theory (e.g., mutual information) and some on cross-correlation (e.g., cross-lagged correlation or windowed cross-lagged correlation). Whereas developers (or users) claim that their algorithms actually measure “synchrony”, there is as yet very little simulation or empirical evidence regarding the validity of synchrony measures, with few exceptions [40,55]. The present study therefore investigated two aspects of the validity of movement-synchrony measures: convergent validity and predictive validity. We applied several algorithms to the same dataset of 30 bivariate time series that represented the motor activity of both the interviewer and interviewee during clinical interviews on somatic complaints. From each interview video, bivariate motion time series were derived. Using these time series, we computed multiple synchronization measures and investigated the correlations between different measures (convergent validity). We also explored which synchrony measure predicted whether the interviewee belonged to the depression group (predictive validity).

### 4.1. Convergent Validity

Regarding the convergent validity, we found that synchrony measures originating from the same algorithmic approach were moderately to highly related. For instance, the three measures of mutual information of the R package mpmi [54] correlated highly among each other. The same was true to a moderate degree for measures of the SUCO algorithm [53], the rMEA package [37], and WCLC–PP [4].

When considering measures originating from different algorithms, their convergent validity (their correlation) varied considerably. The largest correlation was observed between CC–raw and SUCO–ES_abs_ (Spearman *r* = 0.78, *p* < 0.05). Many correlations, however, were insignificant and some were significant and negative, e.g., the correlation between MI–Z and WinCRQA–DET (Spearman *r* = −0.58, *p* < 0.05) or between WCLC–PP–F and WinCRQA–RR (Spearman *r* = −0.46, *p* < 0.05). When analyzing different aspects or facets of synchrony, research should consider synchrony measures of different algorithms instead of different measures of the same algorithm.

In detail, there are differences to other studies. In the study of Schoenherr, Paulick, Worrack, Strauss, Rubel, Schwartz, Deisenhofer, Lutz and Altmann [36], the correlation between rMEA–WCLC and WCLC–PP–F was higher (Pearson *r* = 0.55, *p* < 0.05, see ([36], Table 3, lower left triangle)) than in our study (Pearson *r* = 0.31, not significant). The same holds for the correlation between WinCRQA–RR and WCLC–PP–F (Pearson *r* = 0.769, *p* < 0.05, in ([36], Table 3, lower left triangle) versus *r* = −0.44, *p* < 0.05, in our study). Furthermore, our correlations between different synchrony measures did not correspond with the findings of Luehof [45] and Tschacher and Meier [14]. Depending on the kind of physiological data, Tschacher and Meier [14] found little or no inter-correlations between SUSY–ES_abs_, SUSY–ES_noabs_ and SUCO. Yet in the present study of body movements and their synchronization, these measures correlated to a moderate amount (all Pearson correlations *r* > 0.5, all *p* < 0.05, see Table 2). Luehof [45] investigated body movements in interviews and quantified movement synchrony with CRQA and WCLR–PP. The correlation between CRQA–DET and WCLR–PP–F was *r* = −0.01 ([45] Table 4.40), whereas in the present study WinCRQA–DET and WCLR–PP–F correlated with *r* = −0.41 (*p* < 0.05, see Table 2, lower left triangle). However, it should be noted that in the discussed studies, different parameter settings (e.g., window size) were applied, especially in the recurrence techniques. Therefore, for each algorithm recommendations and guide lines for parameter settings should be developed that can be applied across future studies [40]. Another explanation for the heterogeneity may be that the kind of interaction (interviews versus psychotherapy sessions) and/or the kind of data (cyclic physiological time series versus movement time series characterized by bursts) affect the convergence of synchronization measures. Future studies should therefore test the convergent validity of synchronization measures with multiple and diverse datasets.

Next, we systematized the included synchrony measures using a data-driven approach: Factor analyses suggested two facets of synchrony. Indicators of the first factor were rMEA–WCC, all the variants of CC, and all the measures of SUSY and SUCO. These measures were based on cross-correlations and did not consider a specific time lag between the time series. All the MI measures, rMEA–WCLC, WCLC–PP–F, WCLC–PP–R2, WCLR–PP–F, and WinCRQA–DET loaded on the second factor (when applying a minimum rank factor analysis). MI and WinCRQA–DET are based on information theory and quantify a non-linear relationship in continuous data. The other synchrony measures of this factor use cross-lagged correlations (cross-lagged regression) to quantify a linear relationship between the time series. It should be noted that Schoenherr et al. [36] found a three-factor structure in EFA. The difference to our study may rest in that different synchronization measures were investigated and there was a small dataset in the present study. However, consistent with Schoenherr et al. [36], WCLC–PP–F, WCLR–PP–F, and WinCRQA–DET were assigned to the same factor.

In sum, we agree with Schoenherr et al. [36] by concluding that the convergent validity across the considered algorithmic approaches is insufficient, if present at all. While the mathematical justifications of all the approaches we tested here are clearly given, the quantifications of synchrony they are offering are in most cases only loosely connected. The factor analyses in Schoenherr et al. [36] and in the present study both suggest the presence of multiple facets of synchrony, where one facet appears to summarize coupling in terms of cross-correlation approaches, and the other relates to the frequency of synchronization intervals and the information-theory-based measures.

Further research is needed that can differentiate these synchrony aspects from one another. It would be straightforward to implement large studies with simulated datasets of pairs of time series that represent clear types of coupling between the respective pairs. The coupling may be locally restricted or globally present throughout the time series, coupling may be linear or nonlinear, and time series may be auto-correlated and stationary or not [71]. Such studies can ultimately elucidate which synchrony aspect is recognized by which algorithm. In addition, it would be possible to tailor the parameter settings of the algorithms to serve recognition.

A critical point to discuss is the convergent validity itself. Our study revealed that the absolute value of cross-correlation (CC–abs) was moderately to highly correlated with all the measures of rMEA, SUSY and SUCO (all Pearson *r* > 0.43, see Table 3). Accordingly, these measures formed a separate facet of synchrony in the factor analysis. The cross-correlation is one of simplest measures of synchrony by computing the linear relationship between two time series, not considering any time lag and without segmentation (as in windowed cross-correlations). The benefits of the more sophisticated algorithms rMEA, SUSY, and SUCO lie in the inclusion of surrogate testing that allows the computation of effect sizes and significance even in single-case time series. It remains to be seen how the various correlation-based algorithms fare in heterogeneous and non-stationary data. On the other hand, the measures that assess the frequency of synchronization intervals (WCLC–PP–F and WCLR–PP–F) were related only to cross-recurrence measures (WinCRQA) whereby the signs of correlations were negative (both Pearson *r* ≈ −0.4, see Table 3). The question is whether the validity is given when a measure appears somewhat idiosyncratic; future research should explore in which conditions and in what kind of data the two facets of synchrony may collapse into one factor.

Interestingly, a similar situation regarding convergent validity was present in the measurement of adult attachment [71]. Possibly, the phenomenon of interest itself may have multiple aspects (facets) that are not related in a linear manner and may be measured currently only by one specific instrument (algorithm). Further methodological research is necessary to build bridges between these facets of synchrony, e.g., by developing further instruments (algorithms) or investigating non-linear relationships between the facets of synchrony.

### 4.2. Predictive Validity

Second, we studied the predictive validity based on the assumption that the presence of major depression as well as the degree of symptom load should result in a lower degree of synchrony and fewer synchronization intervals, respectively. In the present naturalistic dataset, more than half of the considered synchronization measures did not correlate with the degree of depressive symptoms, e.g., rMEA–WCLC, all the variants of CC, all the SUSY, and all the WinCRQA measures (see Table 5). The only synchrony measures that corresponded with our hypothesis were WCLC–PP–F and WCLR–PP–F. There was a negative correlation between these synchrony measures and depressive symptoms (Spearman *r*(WCLC–PP–F, PHQ9) = −0.43, *r*(WCLC–PP–F, PHQ9) = −0.47, respectively, both *p* < 0.05). In contrast to our assumption, rMEA–WCC, SUCO–ES–CO, and MI–Z showed positive correlations with depressive symptoms (all Spearman *r* > 0.46, all *p* < 0.05). These synchrony measures suggested that interpersonal interactions with depressed patients are characterized by a higher degree of movement synchrony. These results correspond with [36], who studied predictive validity based on psychotherapy data, finding inconsistent correlations with improvement of interpersonal problems in psychotherapy.

A possible explanation is that the algorithms measure different aspects of movement synchrony, which then correlate differently with depressive symptoms. WCLC–PP–F and WCLR–PP–F measure the frequency of synchronization intervals whereas rMEA–WCC, SUCO–ES–CO, and MI–Z quantify the degree of interrelatedness of both time series. Nevertheless, the present study revealed that in the diagnostic of depression, synchronization measures can lead to contrary conclusions (depressed synchronized less than control versus depressed synchronized more than controls). This raises the problem that the results of different synchrony studies cannot be aggregated when different measures have been used. A solution may be to measure movement synchrony with multiple algorithms, for example, when the relationship between depressive symptoms and synchronization is investigated. This would be comparable to studies on the efficacy of psychological treatment, in which both primary and secondary outcomes are assessed.

### 4.3. Limitations

Our sample of interview videos (bivariate time series) was rather small. Accordingly, the statistical analysis had low statistical power with limited generalizability. Future studies on the validity of synchronization measures should investigate large and diverse samples (e.g., free communication, structured interviews, and psychotherapy sessions) and consider time series related to different behavior modalities (e.g., movement synchrony and facial synchrony) and different contexts (e.g., mirror game and interviews). The present study investigated only movement synchrony in structured interviews.

Previous studies [39,40,41,43] showed in various algorithms that synchronization measures depend on the parameter settings. In the present study, each algorithm was applied with default parameter values recommended by the authors of the algorithms. Possibly, the convergence of synchrony measures depends on equal settings of corresponding parameters. In [36], WinCRQA and WCLR–PP were conducted with a window size of 5 frames (5 s). The correlation of the resulting synchrony measures was *r* = 0.777 ([36], Table 3). In the present study, the window size of WCLR–PP was 125 frames and the window size of WinCRQA was 1500 frames. Both synchrony measures correlated with *r* = −0.41 (see Table 3).

Study designs must be discussed, too. In the present study, we did not control the amount of synchrony in the experimental condition (patients versus controls) so that the “true” synchrony or a proxy for that is not known. Our analysis of predictive validity rested on the assumption that psychopathological symptom load should be linked to movement synchrony during interviews on somatic complaints. There is some plausibility for this assumption; yet it may be also true that both groups of participants were equally synchronized, as the topic of somatic complaints is an engaging topic for depressive as well as healthy interviewees. Additionally, as we discussed previously, the convergent validity of published findings on psychopathology and synchrony is not yet sufficiently robust because these findings originated from differing algorithms and differing parameter settings. Thus, a possible conclusion is that it is too early to study predictive validity; the (convergent) validity of the synchrony measures must be established in the first place.

At the very least, further studies building on the present one are necessary in the field of synchronization research to clarify especially convergent, but also predictive validity. On top of incorporating simulated data with known types of synchronized coupling (in order to analyze convergent validity) [40], experimental data with covert instructions for participants to synchronize (or not) [44,45], and sensitivity analyses on parameter settings and their influences [39,40,43] must be performed.

## 5. Conclusions

To date, only a few comparisons between synchrony measures deriving from different algorithms (frequency-, correlation-, information-based) have been performed systematically. Only recently and in the field of physics have such comparisons been performed on a large scale [72]. In the present study, we pursued a similar goal using a small naturalistic dataset that comprises psychological interaction processes.

Our study revealed that the convergent validity of synchronization measures applied in clinical research range from non-existent to very good. As expected, factor analyses suggested that the different convergence of the measures can be explained by the presence of facets: on the one hand cross-correlation measures and on the other hand measures based on information theory or describing the frequency of synchronization intervals. Moreover, patients with depression and healthy controls can be distinguished by many synchrony measures—which suggests predictive validity. However, some measures suggested that patients and interviewer synchronize less often than dyads with controls, whereas other measures suggested the opposite.

We believe the present study is a promising starting point for addressing the important question of what psychological meaning may reside in synchronization measures. Given the increasing number of synchrony studies in clinical, social, and developmental psychology, these are also pressing open questions in the light of what has been called the “replication crisis” in psychology and medicine.

## Figures and Tables

**Figure 1 entropy-24-01307-f001:**
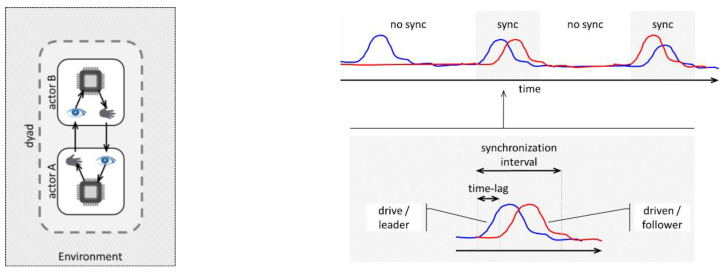
Schematic illustration of a dyad as coupled dynamical system (**left**) and hypothetical motion activity of two interactants with synchronization intervals (**right**).

**Figure 2 entropy-24-01307-f002:**
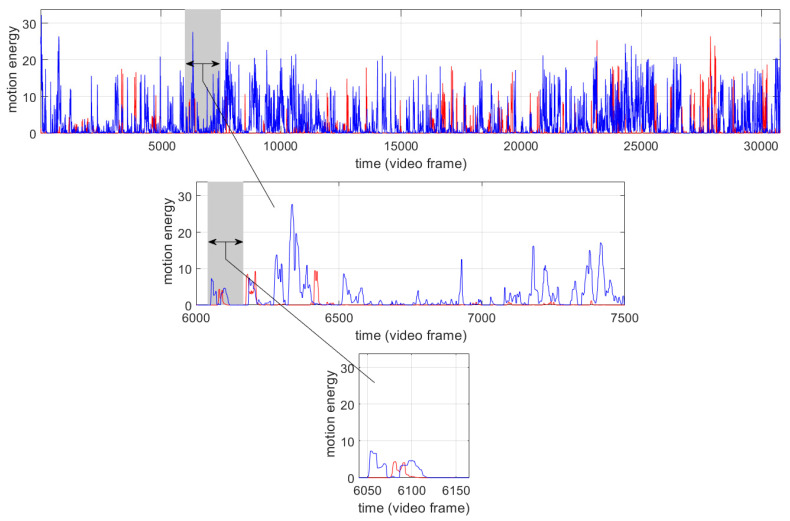
Example of an examined pair of standardized motion-energy time series (participant: red line; interviewer: blue line; 1st row: entire time series; 2nd row: time-series segment with length of 1500 frames; 3rd row: time-series segment with length of 125 frames).

**Table 1 entropy-24-01307-t001:** Description of included study subjects.

	All (*N*_Persons_ = 30)	Healthy Controls (*N*_Persons_ = 15)	Depressive Patients (*N*_Persons_ = 15)	*p*-Value
	Socio-demographic characteristics			
Age in years	25.2 (3.14)	25.5 (3.25)	24.9 (3.10)	0.6091
Gender				1.0000
Male	18 (60.0%)	9 (60.0%)	9 (60.0%)	
Female	12 (40.0%)	6 (40.0%)	6 (40.0%)	
Education				0.1686
No high-school degree	6 (20.0%)	1 (6.67%)	5 (33.3%)	
High-school degree	24 (80.0%)	14 (93.3%)	10 (66.7%)	
Partner status				0.6817
Without partner	22 (73.3%)	10 (66.7%)	12 (80.0%)	
In steady relationship	8 (26.7%)	5 (33.3%)	3 (20.0%)	
	Questionnaires (pre interview)			
Depressive symptoms (PHQ9)	9.43 (7.10)	3.40 (2.44)	15.5 (4.52)	<0.0001
Anxiety symptoms (GAD7)	6.80 (5.76)	1.73 (1.71)	11.9 (3.29)	<0.0001

Note: For continuous variables average and standard deviation are reported and for categorical variables frequency and percentage. For categorical variables a chi-squared or exact Fisher test was applied (the latter, when one or more expected cell frequencies were less than 5). For continuous variables we used a *t*-test or Kruskall–Wallis test (the latter for non-normally distributed data). For more details see [16]. *N* denotes the number of persons.

**Table 2 entropy-24-01307-t002:** Features of synchronization measures.

Algorithm/Package	Method	Global or Local Synchrony Estimation	Time Lag	Significance Test	Sign of Correlations	Peak Picking	Output/Synchrony Scores
CC	Cross- correlation	Global	No	No	Positive and negative values	no	Cross-correlation (CC–raw) Fisher’s Z-transformed CC (CC–Z) Absolut values of CC (CC–abs) R square of CC (CC–R2)
rMEA by Kleinbub and Ramseyer [52]	Windowed cross-lagged correlation, Fisher’s Z-transformed	Local	Yes	Sample shuffling	Absolute values	no	Windowed cross-correlation without time lags (rMEA–WCC) Windowed cross-lagged correlation with time lags (rMEA–WCLC)
SUSY by Tschacher and Haken [53]	Windowed cross-lagged correlation, Fisher’s Z-transformed	Local	Yes	Segment shuffling	Absolute values (SUSY– ES_abs_); positive and negative (SUSY– ES_noabs_)	no	global effect size of absolute values of cross-correlations (SUSY–ES_abs_) global effect size of cross-correlations (SUSY–ES_noabs_)
SUCO by Tschacher and Meier [14]	Windowed regression, Fisher’s Z-transformed	Local	No	Segment shuffling	Absolute values (SUCO– ES_abs_); positive and negative (SUCO– CO, SUCO–ES–CO)	no	global concordance value (SUCO–CO) global effect size of regressions (SUCO–ES_abs_) global effect size of concordance values (SUCO–ES–CO)
WCLC–PP by Altmann [4,12]	Windowed cross-lagged correlation and peak-picking algorithm	Local	Yes	R squared difference test with α = 0.001	Squared (positive)	yes	Frequency of synchrony relative to conversation duration (WCLC–PP–F), average R2 of WCLC within the synchronization intervals (WCLC–PP–R2)
WCLR–PP by Altmann [4,12]	Windowed cross-lagged regression and peak-picking algorithm	Local	Yes	R squared difference test with α = 0.001	Squared (positive)	yes	Frequency of synchrony relative to conversation duration (WCLR–PP–F), average R2 of WCLR within the synchronization intervals (WCLR–PP–R2)
MI by Pardy [54]	Information theory	Global	No	No	not applicable	no	mutual information (MI–raw) Jackknife bias corrected MI (BCMI) (MI–COR) Z-scores of BCMI (MI–Z)
WinCRQA by Coco and Dale [46]	Windowed cross-recurrence	Local	Yes	No	not applicable	no	Recurrence rate in % (WinCRQA–RR) Determination rate in % (WinCRQA–DET) Normalized entropy (WinCRQA–ENTR)

Note: CC: cross-correlation; rMEA: R package motion-energy analysis; SUSY: surrogate synchrony; SUCO: surrogate concordance; WCLC: windowed cross-lagged correlation; WCLR: windowed cross-lagged regression; PP: peak picking; R2: squared correlation; MI: mutual information; WinCRQA: windowed cross-recurrence quantification analysis.

**Table 3 entropy-24-01307-t003:** Pearson correlations (lower left triangle) and Spearman correlations (upper right triangle) of synchronization measures.

	1. CC–raw	2. CC–abs	3. CC–Z	4. CC–R2	5. rMEA–WCC	6. rMEA–WCLC	7. SUSY–ES_abs_	8. SUSY–ES_noabs_	9. SUCO–CI	10. SUCO–ES_abs_	11. SUCO–ES–CI	12. WCLC–PP–F	13. WCLC–PP–R2	14. WCLR–PP–F	15. WCLR–PP–R2	16. MI–raw	17. MI–cor	18. MI–Z	19. WinCRQA–RR	20. WinCRQA–DET	21. WinCRQA–ENTR
1		0.22	1.00 *	0.22	0.48 *	0.22	0.64 *	0.78 *	0.70 *	0.56 *	0.73 *	−0.17	0.04	−0.17	0.06	0.02	−0.01	−0.12	−0.2	0.02	0.18
2	0.69 *		0.22	1.00 *	0.72 *	0.57 *	0.46 *	0.19	0.27	0.25	0.23	−0.25	0.23	−0.26	0.06	0.03	0	0.07	−0.24	−0.18	0.07
3	1.00 *	0.69 *		0.22	0.48 *	0.22	0.64 *	0.78 *	0.70 *	0.56 *	0.73 *	−0.17	0.04	−0.17	0.06	0.02	−0.01	−0.12	−0.2	0.02	0.18
4	0.73 *	0.95 *	0.74 *		0.72 *	0.57 *	0.46 *	0.19	0.27	0.25	0.23	−0.25	0.23	−0.26	0.06	0.03	0	0.07	−0.24	−0.18	0.07
5	0.73 *	0.88 *	0.73 *	0.82 *		0.75 *	0.63 *	0.35	0.50 *	0.56 *	0.53 *	−0.1	0.39 *	−0.16	0.18	0.22	0.18	0.28	−0.32	−0.24	−0.1
6	0.21	0.56 *	0.21	0.44 *	0.68 *		0.33	0.08	0.22	0.33	0.2	0	0.57 *	−0.09	0.09	0.43 *	0.39 *	0.49 *	−0.33	−0.33	−0.1
7	0.72 *	0.66 *	0.72 *	0.64 *	0.74 *	0.40 *		0.57 *	0.52 *	0.54 *	0.54 *	−0.13	0.07	−0.13	0.09	−0.09	−0.12	−0.09	−0.12	0.01	0.17
8	0.74 *	0.43 *	0.74 *	0.50 *	0.40 *	0.02	0.50 *		0.66 *	0.50 *	0.60 *	0	0.12	−0.02	0.16	−0.06	−0.05	−0.28	−0.08	0.22	0.3
9	0.83 *	0.64 *	0.83 *	0.68 *	0.72 *	0.25	0.64 *	0.57 *		0.57 *	0.96 *	−0.12	0.2	−0.07	0.29	−0.02	−0.02	−0.01	−0.02	0.14	0.13
10	0.70 *	0.61 *	0.70 *	0.59 *	0.72 *	0.3	0.64 *	0.51 *	0.72 *		0.62 *	0	0.31	−0.14	0.35	0.05	0.04	0.05	−0.04	0.14	0.22
11	0.82 *	0.62 *	0.82 *	0.64 *	0.73 *	0.21	0.66 *	0.53 *	0.96 *	0.75 *		−0.19	0.14	−0.15	0.2	−0.1	−0.12	−0.03	−0.08	0.14	0.16
12	−0.3	−0.25	−0.3	−0.32	−0.16	0.31	−0.15	−0.01	−0.16	−0.17	−0.29		0.44 *	0.87 *	0.03	0.1	0.06	0.09	−0.46 *	−0.44 *	−0.32
13	0.17	0.32	0.17	0.32	0.38 *	0.65 *	0.15	0.14	0.36	0.26	0.21	0.51 *		0.31	0.48 *	0.29	0.29	0.3	−0.31	−0.27	0.02
14	−0.28	−0.31	−0.28	−0.40 *	−0.21	0.21	−0.15	0.02	−0.11	−0.2	−0.24	0.93 *	0.39 *		0.02	0	−0.01	−0.01	−0.37 *	−0.43 *	−0.28
15	0.14	0.19	0.14	0.21	0.22	0.25	0.08	0.12	0.34	0.3	0.21	0.32	0.66 *	0.35		0.2	0.23	0.07	0.2	0.16	0.22
16	−0.01	−0.06	−0.02	−0.14	0.13	0.53 *	0	−0.16	0.02	0.1	−0.11	0.28	0.49 *	0.26	0.35		0.98 *	0.80 *	−0.18	−0.38 *	−0.17
17	0	−0.07	−0.01	−0.14	0.1	0.47 *	0	−0.16	0.01	0.1	−0.12	0.22	0.46 *	0.22	0.39 *	0.99 *		0.74 *	−0.09	−0.28	−0.15
18	−0.08	−0.05	−0.08	−0.12	0.16	0.58 *	0	−0.26	0.01	0.04	−0.05	0.27	0.44 *	0.27	0.27	0.89 *	0.84 *		−0.3	−0.58 *	−0.25
19	−0.06	−0.15	−0.06	−0.07	−0.24	−0.41 *	−0.09	−0.08	0.03	−0.01	−0.03	−0.44 *	−0.22	−0.31	0.19	−0.16	−0.06	−0.23		0.83 *	−0.05
20	0.16	−0.07	0.16	0.09	−0.19	−0.51 *	0.01	0.2	0.18	0.07	0.17	−0.49 *	−0.24	−0.41 *	0.07	−0.33	−0.22	−0.43 *	0.84 *		0.13
21	0.26	0.23	0.26	0.26	0.1	−0.21	0.28	0.28	0.22	0.37 *	0.28	−0.41 *	−0.12	−0.40 *	0.02	−0.16	−0.15	−0.24	0.04	0.12	

Note: *N*_dyads_ = 30; * *p* < 0.05; CC: cross-correlation; SUSY: surrogate synchrony by Tschacher and Haken [53]; SUCO: surrogate concordance by Tschacher and Meier [14]; rMEA: R package for motion-energy analysis by Kleinbub and Ramseyer [37]; WCLC–PP and WCLR–PP: windowed cross-lagged correlation and windowed cross-lagged regression with subsequent peak picking by Altmann [4,12]; MI: mutual information by Pardy [54]; WinCRQA: windowed cross-recurrence quantification analysis by Coco and Dale [46], and Coco, Mønster, Leonardi, Dale and Wallot [66].

**Table 4 entropy-24-01307-t004:** Loadings of exploratory factor analysis with maximum likelihood estimator (“ML”) and minimum rank factor analysis (“MINRANK”).

	ML		MINRANK	
	Factor 1	Factor 2	Factor 1	Factor 2
CC–raw	**1.00**	0.01	**0.93**	−0.09
CC–abs	**0.70**	−0.05	**0.86**	0.05
CC–Z	**1.00**	0.01	**0.93**	−0.09
CC–R2	**0.74**	−0.12	**0.88**	−0.06
rMEA–WCC	**0.74**	0.15	**0.88**	0.24
rMEA–WCLC	0.23	**0.53**	*0.38*	**0.74**
SUSY–ES_abs_	**0.73**	0.02	**0.80**	0.02
SUSY–ES_noabs_	**0.73**	−0.14	**0.66**	−0.14
SUCO–CI	**0.84**	0.04	**0.88**	0.03
SUCO–ES_abs_	**0.71**	0.11	**0.80**	0.06
SUCO–ES–CI	**0.81**	−0.04	**0.87**	−0.03
WCLC–PP–F	−0.30	0.27	−0.30	**0.68**
WCLC–PP–R2	0.18	**0.49**	0.28	**0.72**
WCLR–PP–F	−0.28	0.26	*−0.31*	**0.62**
WCLR–PP–R2	0.15	0.37	0.24	0.45
MI–raw	−0.00	**1.00**	−0.04	0.81
MI–cor	0.00	**0.99**	−0.04	**0.75**
MI–Z	−0.07	**0.89**	−0.06	**0.82**
WinCRQA–RR	−0.07	−0.15	−0.05	−0.46
WinCRQA–DET	0.16	−0.31	0.12	−0.63
WinCRQA–ENTR	0.26	−0.16	0.34	−0.34
Variance explained by factor	32.8%	18.0%	37.4%	23.2%
Correlation of both factors	−0.02		0.04	
RMSR	0.16		0.12	
RMSEA	0.266		0.301	
TLI	0.367		0.189	

Note: *N*_dyads_ = 30; oblimin rotation; loadings > 0.5 marked bold and cross-loadings > 0.3 italic; CC: cross-correlation; SUSY: surrogate synchrony by Tschacher and Haken [53]; SUCO: surrogate concordance by Tschacher and Meier [14]; rMEA: R package for motion-energy analysis by Kleinbub and Ramseyer [37]; WCLC–PP and WCLR–PP: windowed cross-lagged correlation and windowed cross-lagged regression with subsequent peak picking by Altmann [4,12]; MI: mutual information by Pardy [54]; WinCRQA: windowed cross-recurrence quantification analysis by Coco and Dale [46], Coco, Mønster, Leonardi, Dale and Wallot [66]; RMSR: root mean square of the residuals; RMSEA: root-mean-square error of approximation; TLI: Tucker Lewis Index of factoring reliability.

**Table 5 entropy-24-01307-t005:** Average synchronization depending on group assignment (averages and standard deviations, the *p*-value to Kruskall–Wallis test) and Spearman correlations (*r*) between symptoms and synchrony scores using the entire sample.

	Entire Sample	Healthy Controls	Depressive Patients	Group Comparison		*r* with	*r* with
	*N* = 30	*N* = 15	*N* = 15	*p*-Value		*PHQ9*	*GAD7*
CC–raw	0.02 (0.09)	0.00 (0.06)	0.04 (0.10)	0.1677		0.24	0.39 *
CC–abs	0.06 (0.06)	0.05 (0.03)	0.08 (0.07)	0.1617		0.28	0.39 *
CC–Z	0.02 (0.09)	0.00 (0.06)	0.04 (0.10)	0.1654		0.24	0.39 *
CC–R2	0.01 (0.01)	0.00 (0.00)	0.01 (0.02)	0.1402		0.29	0.39 *
rMEA–WCC	0.11 (0.04)	0.09 (0.03)	0.13 (0.05)	0.0274		0.49 *	0.60 *
rMEA–WCLC	0.09 (0.02)	0.09 (0.02)	0.09 (0.01)	0.1835		0.29	0.36 *
SUSY–ES_abs_	0.59 (1.00)	0.28 (0.92)	0.90 (1.00)	0.0918		0.33	0.43 *
SUSY–ES_noabs_	−2.60 (9.02)	−1.61 (4.66)	−3.59 (12.0)	0.5588		−0.19	−0.02
SUCO–CI	0.39 (0.77)	0.21 (0.62)	0.58 (0.88)	0.1989		0.36	0.43 *
SUCO–ES_abs_	0.97 (1.92)	0.34 (1.25)	1.60 (2.29)	0.0753		0.46 *	0.49 *
SUCO–ES–CI	0.93 (1.48)	0.37 (1.06)	1.48 (1.67)	0.0473		0.49 *	0.56 *
WCLC–PP–F	0.41 (0.11)	0.46 (0.07)	0.36 (0.11)	0.0081		−0.43 *	−0.40 *
WCLC–PP–R2	0.40 (0.02)	0.40 (0.02)	0.40 (0.02)	0.5902		−0.02	−0.02
WCLR–PP–F	0.45 (0.09)	0.49 (0.05)	0.41 (0.11)	0.0114		−0.47 *	−0.39 *
WCLR–PP–R2	0.43 (0.02)	0.43 (0.02)	0.43 (0.03)	0.8538		0.01	0.04
MI–raw	0.70 (0.26)	0.64 (0.27)	0.75 (0.26)	0.2537		0.21	0.23
MI–cor	0.55 (0.21)	0.51 (0.22)	0.60 (0.20)	0.2508		0.20	0.22
MI–Z	64.3 (21.5)	55.3 (17.2)	73.3 (22.1)	0.0197		0.39 *	0.38 *
WinCRQA–RR	41.7 (10.7)	41.4 (12.9)	42.1 (8.32)	0.8581		0.03	−0.06
WinCRQA–DET	99.3 (0.45)	99.3 (0.49)	99.3 (0.42)	0.9594		0.03	−0.06
WinCRQA–ENTR	0.70 (0.02)	0.69 (0.02)	0.70 (0.02)	0.2987		0.19	0.14

Note: * *p* < 0.05; CC: cross-correlation, SUSY: surrogate synchrony by Tschacher and Haken [53]; SUCO: surrogate concordance by Tschacher and Meier [14]; rMEA: R package for motion-energy analysis by Kleinbub and Ramseyer [37]; WCLC–PP and WCLR–PP: windowed cross-lagged correlation and windowed cross-lagged regression with subsequent peak picking by Altmann [4,12]; MI: mutual information by Pardy [54]; WinCRQA: windowed cross-recurrence quantification analysis by Coco and Dale [46], Coco, Mønster, Leonardi, Dale and Wallot [66]; PHQ9: Depression Module of Patient Health Questionnaire; GAD7: Generalized Anxiety Disorder Scale.

## Data Availability

The data that support the findings of this study are available on request from the corresponding author, U.A.

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
