# Peer review of "Cross-Correlation- and Entropy-Based Measures of Movement Synchrony: Non-Convergence of Measures Leads to Different Associations with Depressive Symptoms"

_entropy, 2022, doi:10.3390/e24091307_

Round 1
Reviewer 1 Report
I thank a lot the authors for this wonderful paper! It was very clear to read and results are really important for the field. I thus encourage its publication after minor polishing.
Here are my few comments:
(1) It might be useful and intuitive to add in Table 2 the meaning of different synchronization measures. For example, what do information theory and windowed cross-recurrence capture (as the authors mentioned, the mathematical justifications, etc)?
(2) Did you use the same preprocessing across different synchronization measures to make them maximally comparable?
(3) The p-values are not corrected, despite multiple comparisons. If these analyses are exploratory, please state this explicitly in your introduction and perhaps include this as a potential limitation.
(4) It might be a good idea to clarify why max lag = 5 s (not shorter or longer) was used.
Author Response
(1) It might be useful and intuitive to add in Table 2 the meaning of different synchronization measures. For example, what do information theory and windowed cross-recurrence capture (as the authors mentioned, the mathematical justifications, etc)?
- We added much as possible information about the different synchrony measures. However, describing the meaning of measure was little bit difficult because the most authors of algorithm state that their algorithm measure synchrony. Therefore we added the last column in with the outcome measures described in short.
(2) Did you use the same preprocessing across different synchronization measures to make them maximally comparable?
- Of course, all algorithms analyzed the same time series pairs. One exception is the cross recurrence quantification analysis. Here, the time series transformed to the unit interval [0, 1]. This is part of the algorithm to identify whether both persons are in the same state. This is highlight in the method part.
(3) The p-values are not corrected, despite multiple comparisons. If these analyses are exploratory, please state this explicitly in your introduction and perhaps include this as a potential limitation.
- In the revised manuscript we highlighted in the introduction and discussion section that the present study is an explorative study. However, we did not correct the p-values when comparing patients and controls. In our study we wanted to show that some synchrony measures point in this direction (e.g. less synchrony in depressed patients) and other measures point in another direction (e.g. more synchrony in depressed patients). It was not the primary aim to proof whether depressed synchronize more often than controls. If we apply a Bonferroni correction then no variable would be significant when comparing both groups.
(4) It might be a good idea to clarify why max lag = 5 s (not shorter or longer) was used.
- As described in the method section, we applied the parameter settings (e.g. max lag = 5 s) as the authors of algorithms recommend. By our knowledge, the max leg parameter vary between 2.5 (Altmann, 2011) and 7 sec (Robinson, Herman, & Kaplan, 1982). max lag = 5 sec is often applied in psychotherapy research, e.g. by Altmann et al., 2019; Paulick, Deisenhofer, et al., 2018; Ramseyer & Tschacher, 2011; Schoenherr, Paulick, et al., 2019). It should be noted that max lag limits the range in which both interacting persons can synchronize. Schoenherr, Strauss et al. (2019) reported that the average time lag when synchronizing was M=2.55 sec which is much small than the max lag parameter.
Reviewer 2 Report
The presented article deals with a very interesting problem from both a medical and a methodological point of view. This is a current topic, especially in today's post-covid era, when there has been an increase in psychological problems in people, depression and anxiety.
Although the content of the article is interesting and very up-to-date, from the point of view of the current form, it presents numerous pitfalls for the reader. Since the use of this study can have a great impact in practice, I recommend its fundamental modification. The article works with a large number of methods and due to its current notation form, it seems confusing. The mathematical foundations and assumptions of the methods are not formulated. Verification of the assumptions of the methods is not carried out or stated. This can result in misleading understanding or results. Orientation in the text and results is therefore difficult and verification of the relevance of the results is very complex.
The article provides a lot of information, a lot of citations, and works with a lot of abbreviations. Citations of professional literature are often very long with a large number of authors, which makes the text (in the entire article) more difficult to read. It would be appropriate to introduce an abbreviated citation, e.g. Delaherche et al [10] instead of listing all authors. From the point of view of the number of abbreviations, I recommend authors to include an overview table with definitions of abbreviations or with a reference to the literature of the methodology that the abbreviations denote. In some cases, the abbreviation is not defined (see Ch. 1.2 WCLC, WCLR). Also due to the large number of methods, I recommend structuring the article and creating subsections for methodological groups.
I recommend editing Chapter 1.3. with research questions and adding information what is the novelty to the article compared to existing studies. Even from the point of view of citing the publication, it is advisable for the reader to make this information easily identifiable in the article.
Further, I also recommend editing and supplementing Chapter 2.2 The table of descriptive statistics is appropriate, but it is necessary to supplement and unify the information on which time series will be used for analyses. I completely miss the time series graphs and the description of their timeline. In the text with the results, it is difficult to navigate how large time series or segments are included in the analysis. Whether series length 30 observations, smaller or larger ranges. At the same time, the ranges (sample size) of just such small lengths fundamentally affect the quality of estimates, they can lead to misleading results and consequently to the use of robust statistics and adjusted indicators. Also missing is information about the nature of the input data; they are stationary, non-stationary, with structural breaks, heteroscedastic, noisy due to the used technology of acquisition of images, assumed level of noise, etc.
I would also suggest a separate chapter for the authors to describe the experiment that led to the acquisition of the input time series. Since a possible replication of a study inspired by this study can be expected, good documentation is important.
I find the recording of the methodology used to be insufficient. Therefore, I recommend supplementing the description of mathematical models and formulas, including the description of variables, defining the index of determination adjusted with regard to the sample size. In the results section, model estimates including quality measures of the estimated parameters and the model as a whole. It would also be appropriate to supplement the graphs with autocorrelation (leads and lag) in the event that the memory of the process or significant component at non-zero displacement. I also recommend adding a reference to the literature for the PP algorithm (pp. 20) and Shannon's information (pp. 21). It is also appropriate to add information on the kernel smoothing used. In such sample size it can be expected edge effect in some kernel estimator.
In the case of Chapter 3.1, it would also be appropriate to supplement the mathematical formulation of the problem. If I understand correctly, then it is essentially a two-stage correlation. Again, it is not clear what lengths of time series are correlated. For reflection and commentary in the publication, I present to the authors the possibility of using partial correlations or time series co-movement techniques. The question is whether dynamic correlation or time-frequency co-spectral analysis should be considered based on the nature of the input data.
Finally, I recommend editing the conclusion. Its form is formulated rather to state the current state and outlook for further research. Here, the authors should briefly summarize the results achieved, the answers to the research questions and emphasize whether and what novelty they have achieved in the article.

Author Response
Although the content of the article is interesting and very up-to-date, from the point of view of the current form, it presents numerous pitfalls for the reader. Since the use of this study can have a great impact in practice, I recommend its fundamental modification. The article works with a large number of methods and due to its current notation form, it seems confusing. The mathematical foundations and assumptions of the methods are not formulated. Verification of the assumptions of the methods is not carried out or stated. This can result in misleading understanding or results. Orientation in the text and results is therefore difficult and verification of the relevance of the results is very complex.
- We revised the description of algorithms in the method section. However, there is a space limitation, so the our description is short, of course. We cited the papers in which the algorithms are described in detail (including their mathematical foundation). We hope that this is a good compromise.
The article provides a lot of information, a lot of citations, and works with a lot of abbreviations. Citations of professional literature are often very long with a large number of authors, which makes the text (in the entire article) more difficult to read. It would be appropriate to introduce an abbreviated citation, e.g. Delaherche et al [10] instead of listing all authors. From the point of view of the number of abbreviations, I recommend authors to include an overview table with definitions of abbreviations or with a reference to the literature of the methodology that the abbreviations denote. In some cases, the abbreviation is not defined (see Ch. 1.2 WCLC, WCLR). Also due to the large number of methods, I recommend structuring the article and creating subsections for methodological groups.
- We revised Table 2 which gives an overview about the algorithms. Due to your hint, we added the explanations for the abbreviations.
- We changed the citation. Now, in the most cases the numeric format, e.g. [10], or the short form of authors, e.g. Delaherche et al. [10], is used.
I recommend editing Chapter 1.3. with research questions and adding information what is the novelty to the article compared to existing studies. Even from the point of view of citing the publication, it is advisable for the reader to make this information easily identifiable in the article.
- Thank for the hint. We revised subsection 1.3, highlighted the explorative character of our study and described the novelty of our study.
Further, I also recommend editing and supplementing Chapter 2.2 The table of descriptive statistics is appropriate, but it is necessary to supplement and unify the information on which time series will be used for analyses. I completely miss the time series graphs and the description of their timeline. In the text with the results, it is difficult to navigate how large time series or segments are included in the analysis. Whether series length 30 observations, smaller or larger ranges. At the same time, the ranges (sample size) of just such small lengths fundamentally affect the quality of estimates, they can lead to misleading results and consequently to the use of robust statistics and adjusted indicators. Also missing is information about the nature of the input data; they are stationary, non-stationary, with structural breaks, heteroscedastic, noisy due to the used technology of acquisition of images, assumed level of noise, etc.
- Thank you for this important advice. Now, the analysis steps (video recording, motion capture, computation of multiple synchrony measures based on motion time series, analysis of synchrony scores) are better described.
- We also included a plot with time series. The time series are non-stationary and non-normal distributed as described in the manuscript. In brief, the measurement of body motion with motion energy analysis as well as noise filtering by moving median are reported in subsection 2.5.
- Due to the non-normal distributed synchrony scores we used robust statistics, e.g. Spearman correlation and minimum rank factor analysis.
I would also suggest a separate chapter for the authors to describe the experiment that led to the acquisition of the input time series. Since a possible replication of a study inspired by this study can be expected, good documentation is important.
- Please note that the presented analysis is a secondary analysis. Therefore the experiment is described in subsection 2.4 only in brief. More details reported in the cited papers of primary study: [16,47,48].
I find the recording of the methodology used to be insufficient. Therefore, I recommend supplementing the description of mathematical models and formulas, including the description of variables, defining the index of determination adjusted with regard to the sample size.
- We applied the usual methods when studying the validity of synchrony measures, e.g. correlations and factor analysis. The mathematical models behind factor analysis are well-known in the community.
In the results section, model estimates including quality measures of the estimated parameters and the model as a whole.
- We added indices describing the model fit of factor analyses, see Table 4.
It would also be appropriate to supplement the graphs with autocorrelation (leads and lag) in the event that the memory of the process or significant component at non-zero displacement.
- Did not add such graphs. Motion energy time series and their characteristics are well known in clinical research. However, the handling of auto-correlation is discussed in the research community. The presence of auto-correlation and motivated e.g. the development of windowed cross-lagged regression, see subsection 2.5.4. Other algorithms “ignore” auto-correlation.
I also recommend adding a reference to the literature for the PP algorithm (pp. 20) and Shannon's information (pp. 21).
- References added.
It is also appropriate to add information on the kernel smoothing used. In such sample size it can be expected edge effect in some kernel estimator.
- Please note that the kernel smoothing is applied on density function when estimating the mutual information of time series. It is not applied on the N=30 dyads. We added a reference in which more information is provided about the mutual information computation.
In the case of Chapter 3.1, it would also be appropriate to supplement the mathematical formulation of the problem. If I understand correctly, then it is essentially a two-stage correlation. Again, it is not clear what lengths of time series are correlated. For reflection and commentary in the publication, I present to the authors the possibility of using partial correlations or time series co-movement techniques. The question is whether dynamic correlation or time-frequency co-spectral analysis should be considered based on the nature of the input data.
- Maybe there is a misunderstanding. In the present study, we addressed two research questions, one on the convergent validity and one on the predictive validity of synchrony measures. For each research question we applied multiple methods, e.g. a correlation analysis and factor analysis to investigate the convergent validity. In the revised manuscript, we described the steps of study more detailed and at several places in the manuscript.
Finally, I recommend editing the conclusion. Its form is formulated rather to state the current state and outlook for further research. Here, the authors should briefly summarize the results achieved, the answers to the research questions and emphasize whether and what novelty they have achieved in the article.
- Thank you for the helpful hint. We revised the discussion section, accordingly. We added a summary of results, point out consistency with other studies and discuss questions and methodology of future studies, for example. Furthermore we extended the limitations subsection.
Round 2
Reviewer 2 Report
After revising the manuscript, I still feel there is I see space for additions and improvements which makes the article clearer and strengthens its citation potential, see the attached document.
